# A putative origin of the insect chemosensory receptor superfamily in the last common eukaryotic ancestor

Richard Benton[1]*, Christophe Dessimoz[1,2,3,4,5], David Moi[1,2,3]

[1]Center for Integrative Genomics, Faculty of Biology and Medicine, University of Lausanne, Lausanne, Switzerland; [2]Department of Computational Biology, Faculty of Biology and Medicine, University of Lausanne, Lausanne, Switzerland; [3]Swiss Institute of Bioinformatics, Lausanne, Switzerland; [4]Department of Genetics, Evolution and Environment, University College London, London, United Kingdom; [5]Department of Computer Science, University College London, London, United Kingdom

**Abstract** The insect chemosensory repertoires of Odorant Receptors (ORs) and Gustatory Receptors (GRs) together represent one of the largest families of ligand-gated ion channels. Previous analyses have identified homologous 'Gustatory Receptor-Like' (GRL) proteins across Animalia, but the evolutionary origin of this novel class of ion channels is unknown. We describe a survey of unicellular eukaryotic genomes for GRLs, identifying several candidates in fungi, protists and algae that contain many structural features characteristic of animal GRLs. The existence of these proteins in unicellular eukaryotes, together with *ab initio* protein structure predictions, provide evidence for homology between GRLs and a family of uncharacterized plant proteins containing the DUF3537 domain. Together, our analyses suggest an origin of this protein superfamily in the last common eukaryotic ancestor.

*For correspondence:
Richard.Benton@unil.ch

## Introduction

The insect chemosensory receptor superfamily, comprising Odorant Receptors (ORs) and Gustatory Receptors (GRs), forms a critical molecular interface between diverse chemical signals in the environment and neural activity patterns that evoke behavioral responses (*Benton, 2015*; *Joseph and Carlson, 2015*; *Robertson, 2019*; *Rytz et al., 2013*; *van Giesen and Garrity, 2017*). Most insect genomes encode dozens to hundreds of different, often species-specific, ORs and/or GRs. Detailed analyses, in particular in *Drosophila melanogaster,* indicate that the vast majority of these are likely to be expressed in, and define the chemical response properties of, distinct subpopulations of peripheral sensory neurons (*Chen and Dahanukar, 2020*; *Joseph and Carlson, 2015*; *Scott, 2018*; *Vosshall and Stocker, 2007*).

Insect ORs and GRs – the former having derived from an ancestral GR (*Robertson, 2019*; *Robertson et al., 2003*) – contain seven transmembrane (TM) domains (*Clyne et al., 2000*; *Clyne et al., 1999*; *Scott et al., 2001*; *Vosshall et al., 1999*). In contrast to vertebrate olfactory and taste receptors, which belong to the G protein-coupled receptor (GPCR) superfamily of seven TM domain proteins (*Glezer and Malnic, 2019*; *Yarmolinsky et al., 2009*), insect ORs and GRs have the opposite topology, with an intracellular N-terminus (*Benton et al., 2006*; *Lundin et al., 2007*). Functional analyses of these proteins in heterologous expression systems indicate that they form ligand-gated ion channels (*Butterwick et al., 2018*; *Sato et al., 2008*; *Sato et al., 2011*; *Wicher et al., 2008*). Insect ORs assemble into heteromeric (probably tetrameric) complexes likely composed of two subunits each of a tuning OR, which recognizes odor ligands, and a universal co-receptor,

ORCO, which is critical for complex assembly, subcellular trafficking, and – together with the tuning OR – forms the ion conduction pore (*Benton et al., 2006*; *Butterwick et al., 2018*; *Larsson et al., 2004*; *Sato et al., 2008*; *Wicher et al., 2008*). GRs are less well-characterized but are also likely to function in multimeric complexes of one or more different subunits (*Joseph and Carlson, 2015*; *Scott, 2018*). A cryogenic electron microscopy (cryo-EM) structure of an ORCO homotetramer (*Butterwick et al., 2018*) – which can conduct ions itself upon stimulation with artificial ligands (*Jones et al., 2011*) – demonstrated that this receptor adopts a novel fold unrelated to any known family of ion channels. Analysis of amino acid conservation across the OR repertoire (*Butterwick et al., 2018*) and de novo structure predictions of tuning ORs, guided by patterns of amino acid co-evolution (*Hopf et al., 2015*), suggest that this fold is globally similar for all ORs (and potentially GRs).

The unusual nature of these proteins has prompted significant interest to understand their evolution. The most extensive comparative genomic analyses have focused on Insecta, tracing the origins of the OR family from the ancestral insect GRs (*Brand et al., 2018*; *Missbach et al., 2014*). Surveys for GR-like (GRL) proteins beyond insects have identified members of this family across Protostomia (including in Annelida, Nematoda, and Mollusca) as well as in a limited number of Deuterostomia (including Echinodermata and Hemichordata, but not Chordata) (*Eyun et al., 2017*; *Robertson, 2015*; *Saina et al., 2015*). Several non-bilaterian animals have recognizable GRLs, including Cnidaria and Placozoa (*Eyun et al., 2017*; *Nordstrom et al., 2011*; *Robertson, 2015*; *Saina et al., 2015*). Although only very sparse expression and functional information exists outside Insecta, several lines of evidence indicate that members of this superfamily have roles beyond chemosensation. For example, two homologs in *Caenorhabditis elegans* (LITE-1 and GUR-3) function in light detection (*Edwards et al., 2008*), either as a photoreceptor (*Gong et al., 2016*), or indirectly through recognition of cellular chemical products produced upon light exposure (*Bhatla and Horvitz, 2015*). (Isoforms of the *D. melanogaster* LITE-1 ortholog, GR28b, also have other sensory roles, notably in thermosensation and light-sensing (*Ni et al., 2013*; *Xiang et al., 2010*).) Another *C. elegans* homolog functions in motoneurons to control egg-laying (*Moresco and Koelle, 2004*). GRLs in the sea anemone *Nematostella vectensis* and the purple sea urchin, *Strongylocentrotus purpuratus* are expressed early during development (*Saina et al., 2015*), and one of the *N. vectensis* proteins may have a role in apical body patterning (*Saina et al., 2015*).

Although this family of established (or presumed) ion channels represents one of the largest and most functionally diverse in nature, its evolutionary origin remains unknown. We previously suggested potential homology between GRLs and a family of uncharacterized plant proteins containing the Domain of Unknown Function 3537 (DUF3537), based upon their predicted seven TM domains and intracellular N-termini (*Benton, 2015*). However, this proposal was questioned (*Robertson, 2019*) because DUF3537 proteins lack other features that characterize animal GRLs, such as a motif in TM7 and conserved introns near the 3′ end of the corresponding gene (*Robertson, 2015*; *Robertson, 2019*; *Saina et al., 2015*). Moreover, if insect ORs/GRs and plant DUF3537 proteins were derived from a common ancestor, we might expect to find related proteins encoded in the genomes of unicellular eukaryotes. This study aimed to profit from the wealth of genomic information now available to investigate the potential existence of GRL homologs in such species to further trace the birth of this remarkable protein family.

## Results

### Screening and assessment of candidate GRLs in unicellular eukaryotes

We used diverse insect ORs and GRs, other animal GRLs and plant DUF3537 families as sequence queries in BLAST searches of protein and genomic sequence databases of unicellular organisms (see Materials and methods). Significant hits were subjected to further assessment to exclude spurious similarities, retaining those that fulfilled most or all of the following criteria: (i) reciprocal BLAST using a candidate sequence as query identified a known GRL or DUF3537 member as a top hit (or no significant similarity to other protein families); (ii) ~ 350–500 amino acids long, similar to GRLs/DUF3537 proteins; (iii) predicted seven TM domains; (iv) intracellular N-terminus; (v) longer intracellular loops than extracellular loops (a feature of the insect receptors [*Otaki and Yamamoto, 2003*;

*Robertson, 2015*]). These analyses identified 17 sequences from Fungi, Protista and unicellular Plantae (*Table 1* and *Figure 1*), described in more detail below.

To further assess these candidate homologs (which we refer to as GRLs hereafter), we used their sequences to build and compare Hidden Markov Models (HMMs) using HHblits (*Remmert et al., 2012*), a remote homology detection tool that is more sensitive than BLAST (*Steinegger et al., 2019*) (see Materials and methods). We constructed HMMs for candidate GRLs, as well as a representative set of animal GRLs and DUF3537 proteins. Each HMM was used as a query to perform all-versus-all alignments. A similarity matrix comparing these alignments was compiled by parsing the probabilities of each alignment (*Figure 1—figure supplement 1*). As expected, alignments of HMMs seeded by animal and plant proteins each form clusters of high probability similarity, although we also detected similarity between these clusters, indicative of homology. Importantly, HMMs of all new candidate sequences display significant similarity to those of multiple animal and/or plant proteins. Some candidates clustered more closely with the animal sequences, while others displayed similar probabilities with both animal and plant proteins, an observation consistent with phylogenetic analyses presented below.

We also examined the candidate homologs for the only known primary sequence feature of animal GRLs, a short motif located in the C-terminal half of TM7: (T/S)Yhhhhh(Q/K/E)(F/L/M), where h denotes a hydrophobic amino acid (*Robertson, 2015*). This motif is diagnostic, but not definitive: many insect tuning ORs (as well as some GRs/GRLs) have divergent amino acids at some or all four positions (*Robertson, 2019*; *Scott et al., 2001*). Structural and functional analyses of a subset of residues of this motif in ORCO indicates that the TY residues form part of the interaction interface between subunits (*Figure 2A*), and that their mutation is detrimental to function in some, but not all, combinations of subunits (*Butterwick et al., 2018*; *Nakagawa et al., 2012*). The terminal L residue of the motif is part of the channel gate (*Figure 2A*), and its mutation alters ion permeation selectivity (*Butterwick et al., 2018*). These observations suggest that divergence from the GRL motif in a given

**Table 1.** Candidate GRLs in unicellular eukaryotes.
Protein sequences are provided in *Supplementary file 1*. Protein nomenclature is provisional and does not imply orthology between species.

| Kingdom | Phylum | Species | Isolate | Alternative name | Common name | Provisional protein name | Accession/version |
|---|---|---|---|---|---|---|---|
| Fungi | Chytridiomycota | *Spizellomyces punctatus* | DAOM BR117 | | chytrid fungus | SpunGRL1 | XP_016607089.1 |
| | | *Spizellomyces palustris* | CBS 455.65 | *Phlyctochytrium palustre* | chytrid fungus | SpalGRL1 | TPX68946.1 |
| Protista | Amoebozoa | *Protostelium aurantium var. fungivorum* | - | *Planoprotostelium fungivorum* | - | PfunGRL1 | PRP89608.1 |
| | Apusozoa | *Thecamonas trahens* | ATCC 50062 | *Amastigomonas trahens* | zooflagellate | TtraGRL1 | XP_013761079.1 |
| | | | | | | TtraGRL2 | XP_013753662.1 |
| | | | | | | TtraGRL3 | XP_013759733.1 (trimmed) |
| | | | | | | TtraGRL4 | XP_013759396.1 |
| | | | | | | TtraGRL5 | XP_013757274.1 |
| | | | | | | TtraGRL6 | XP_013755387.1 |
| | *Incertae sedis*/Chromerdia (superphylum: Alveolata) | *Vitrella brassicaformis* | CCMP3315 | - | chromerid | VbraGRL1 | CEM13019.1 |
| | | | | | | VbraGRL2 | CEL93132.1 |
| | | | | | | VbraGRL3 | CEM19221.1 |
| | | | | | | VbraGRL4 | CEM01650.1 |
| | | | | | | VbraGRL5 | CEM10760.1 |
| | | | | | | VbraGRL6 | CEM25255.1 |
| Plantae | Chlorophyta | *Chloropicon primus* | - | | | CpriGRL1 | QDZ19318.1 |
| | | *Micromonas pusilla* | CCMP1545 | *Chromulina pusilla* | | MpusGRL1 | XP_003054778.1 |

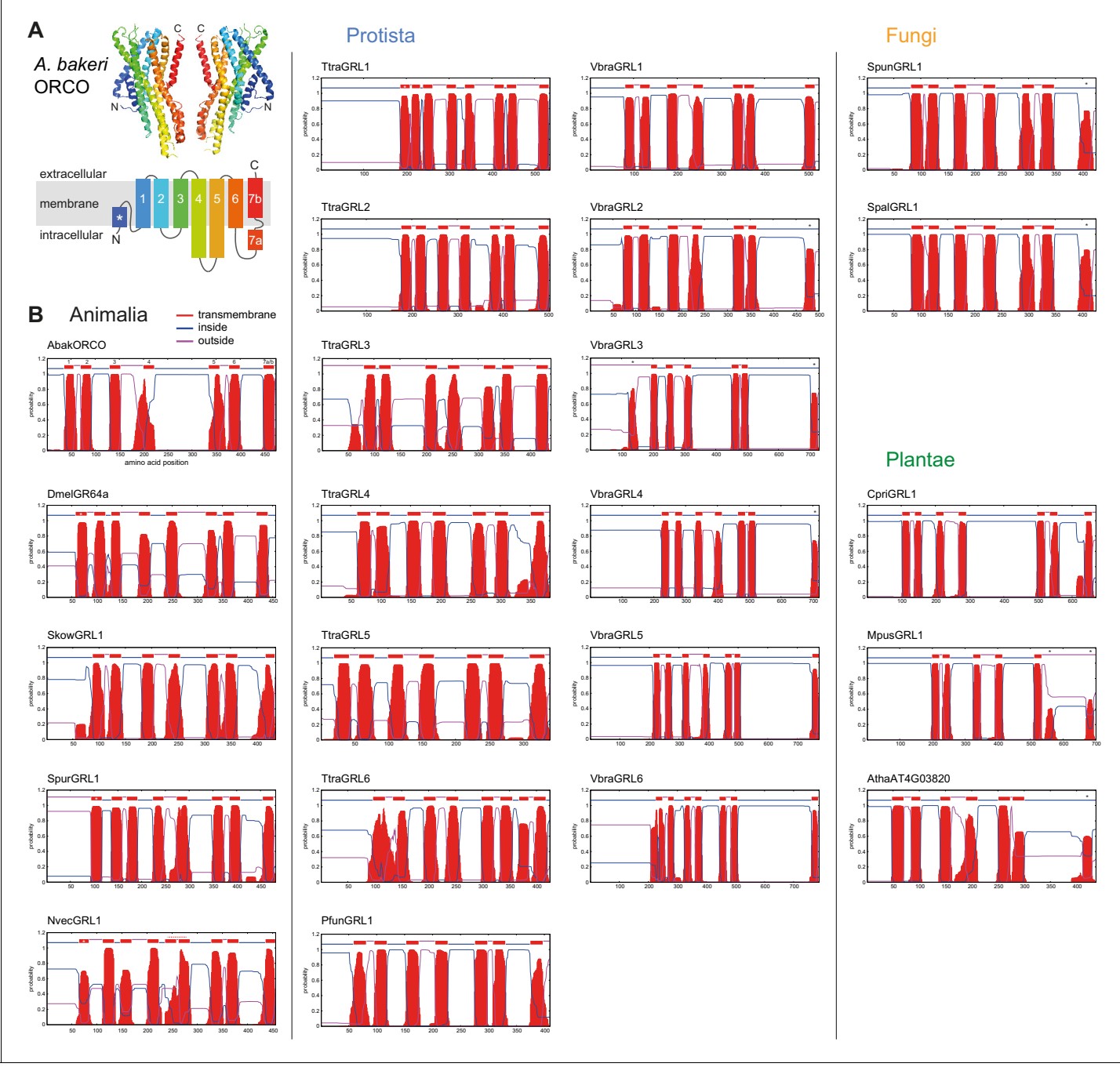

**Figure 1.** Transmembrane topology predictions of GRLs. (A) Top: cryo-EM structure of *Apocrypta bakeri* ORCO (AbakORCO) (PDB 6C70 [*Butterwick et al., 2018*]); only two subunits of the homotetrameric structure are visualized. Bottom: Schematic of the membrane topology of AbakORCO (adapted from *Butterwick et al., 2018*), colored as in the cryo-EM structure. The white asterisk marks a helical segment that forms part of a membrane re-entrant loop in the N-terminal region. TM domain seven is divided into a cytoplasmic segment (7a) and a membrane-spanning segment (7b). (B) TM domain and topology predictions of the previously described and newly-recognized GRLs and DUF3537 proteins (Dmel, *Drosophila melanogaster*; Skow, *Saccoglossus kowalevskii*; Spur, *Strongylocentrotus purpuratus*; Nvec, *Nematostella vectensis*; Atha, *Arabidopsis thaliana*; see *Table 1* for other species abbreviations and sequence accessions). Each plot represents the posterior probabilities of transmembrane helix and inside/outside cellular location along the protein sequence, adapted from the output of TMHMM Server v2 (*Krogh et al., 2001*). In several sequences an extra transmembrane segment near the N-terminus is predicted (marked by a white asterisk in the N-best prediction above the plot); this may represent the re-entrant loop helical region observed in ORCO, rather than a transmembrane region; in at least one case (SpurGRL1) the designation of this region as a TM domain, leads to an atypical (and presumably incorrect) prediction of an extracellular N-terminus. Conversely, in a subset of proteins individual TM domains are not predicted (notably TM7, black asterisks above the N-best plot), which is likely due to subthreshold predictions for TM domainsin

*Figure 1 continued on next page*

*Figure 1 continued*

these regions. In NvecGRL1, the long TM4 helix (which projects into the intracellular space in ORCO [*Butterwick et al., 2018*]) is mis-predicted as two TM domains (dashed red line). Independent membrane topology predictions for unicellular species' GRLs were obtained using TOPCONs (*Supplementary file 2*), with largely consistent results.

The online version of this article includes the following figure supplement(s) for figure 1:

**Figure supplement 1.** Probabilities of alignments of HMMs of known and candidate GRLs and DUF3537 proteins.

protein could be compatible with a conserved function as an ion channel, albeit with different complex assembly and biophysical properties. Bearing these observations in mind, candidate GRLs were inspected for this motif, as described below (*Figure 2B*). Finally, the corresponding genes were examined for the existence of the 3' introns characteristic of the animal genes (*Robertson, 2015*; *Saina et al., 2015*); this analysis was ultimately uninformative because of the scarcity of introns in most of these organisms (*Roy and Gilbert, 2006*; *Figure 2B*).

## Candidate GRLs from unicellular eukaryotes

The fungal kingdom is thought to be the closest relative to Animalia (*Baldauf and Palmer, 1993*). A single candidate GRL was identified in two fungi, *Spizellomyces punctatus* (*Russ et al., 2016*) and *Spizellomyces palustris* (*van de Vossenberg et al., 2019*). The fungal proteins exhibit the secondary structural features of GRLs (*Figure 1*), but only one of the four positions of the TM7 motif is conserved (*Figure 2B*). A 3' intron is not in the same position of the characteristic last intron of animal GRL genes (*Figure 2B*). Both of these species are chytrids, an early diverging lineage of fungi that retains some features of the last common opisthokont ancestor of animals and fungi (*Medina and Buchler, 2020*). Chytrids have diverse lifestyles, but are notable for their reproduction via zoospores, which use a motile cilium to swim or crawl.

Taxonomic classification of many single-celled eukaryotes remains unresolved, and we use the term Protista to cover all unicellular species that are neither Fungi nor Plantae. Three such species were found to encode GRLs. The marine gliding zooflagellate *Thecamonas trahens* (*Cavalier-Smith and Chao, 2010*; *Howe et al., 2020*) has six candidate proteins. Beyond secondary structural similarity (*Figure 1*), the proteins have up to three conserved residues of the TM7 motif (if counting a Y→F substitution in the second position as conservative, as observed in some animal GRLs (e.g., SpurGRL1)) (*Figure 2B*). The chromerid *Vitrella brassicaformis* (*Woo et al., 2015*), a free-living, non-parasitic photosynthetic protist, also has six GRLs, most of which have two conserved positions within the TM7 motif. Finally, the amoebozoan *Protostelium aurantium var. fungivorum* (*Hillmann et al., 2018*), has a single GRL. Protosteloid amoebae differ from dictyostelids by producing simple fruiting bodies with only one or few single stalked spores.

DUF3537 domain proteins are widely (and possibly universally) encoded in higher plant genomes, typically comprising small families of 4–12 members (*Benton, 2015*). Single proteins were also found in unicellular plants ('green algae'), including the marine microalgae *Chloropicon primus* (*Lemieux et al., 2019*) and *Micromonas pusilla* (*van Baren et al., 2016*; *Worden et al., 2009*). As for higher plant sequences, the TM7 motif is largely unrecognizable. The relationship between DUF3537 proteins and the GRL superfamily may have been overlooked earlier because protein alignments are impeded by the longer length of several intracellular loops (IL) of the plant proteins, notably IL3 in all DUF3537 proteins, and IL2 in the green algal proteins (~200 residues in the *C. primus* homolog) (*Figure 1B*). We note that ORCO is also distinguished from other insect ORs and GRs by an additional ~60–70 amino acids in IL2 (*Benton et al., 2006*), a region that contributes to channel regulation (*Bahk and Jones, 2016*; *Mukunda et al., 2014*).

## Phylogenetic analysis of candidate GRLs

The candidate GRLs are extremely divergent in primary sequence: pairwise alignment of the new proposed family members, together with representative animal and plant proteins, reveal as little as 10% amino acid identity. While this divergence does not preclude their definition as homologs – insect OR families themselves have an average of only ~20% amino acid identity (*Butterwick et al., 2018*) – it makes it difficult to infer homology from sequence alone, and hinders confident multiprotein alignment. Indeed, an alignment of these 17 sequences, together with selected animal GRLs

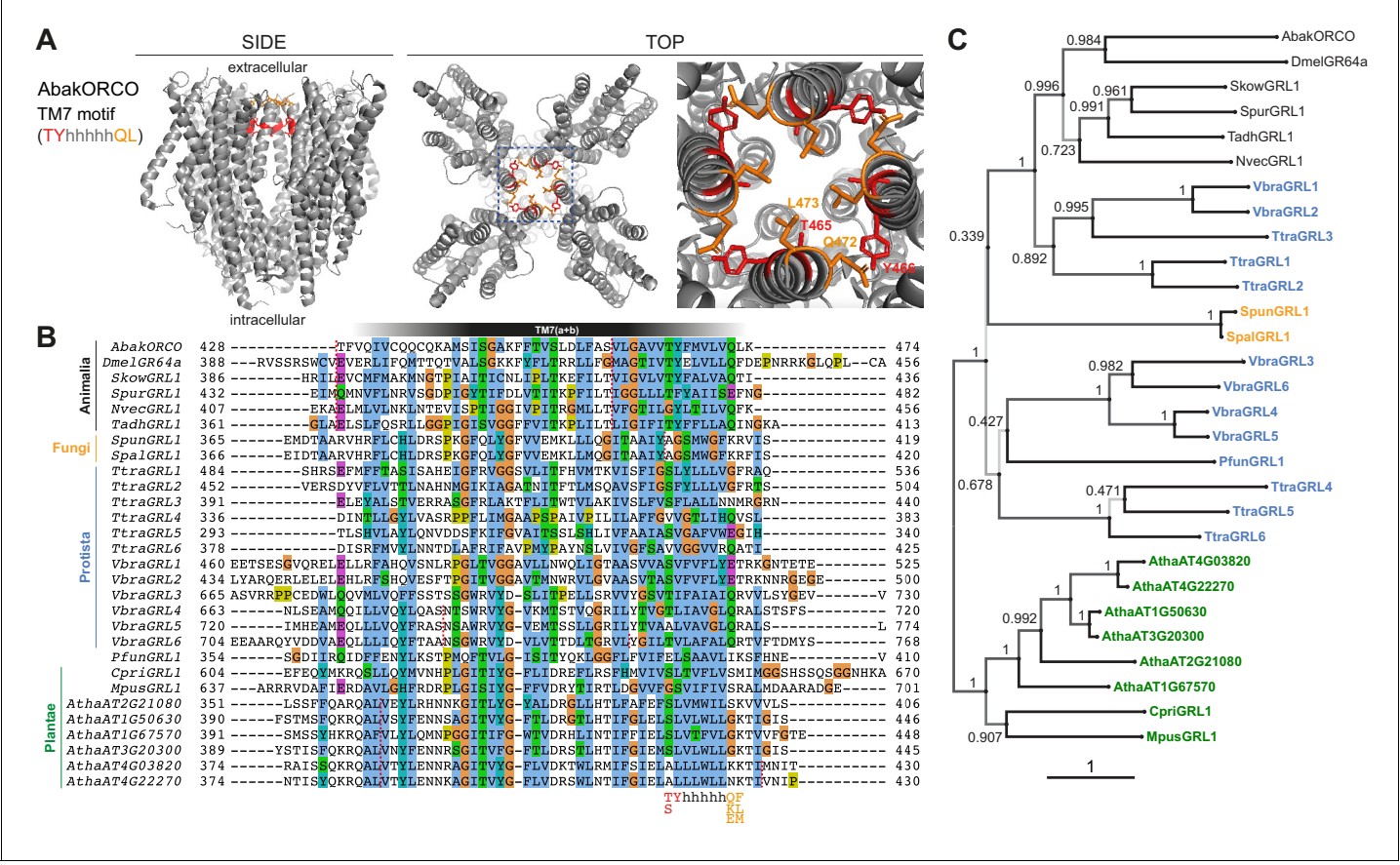

**Figure 2.** Conservation and divergence in TM7 features and GRL phylogeny. (**A**) Side and top views of the cryo-EM structure of the ORCO homotetramer (*Butterwick et al., 2018*), in which the TM7 motif amino acid side chains are shown in stick format and colored red or orange. The region in the dashed blue box, representing the extracellular entrance to the ion channel pore, is shown in a magnified view on the far right. (**B**) Multiple sequence alignment of the C-terminal region (encompassing TM7) of unicellular eukaryotic GRLs and selected animal GRLs and plant DUF3537 proteins. Tadh, *Trichoplax adhaerens*; other species abbreviations are defined in *Figure 1* and *Table 1*. The TM7 motif consensus amino acids (and conservative substitutions) are indicated below the alignment; h indicates a hydrophobic amino acid. Red dashed lines on the alignment indicate positions of predicted introns within the corresponding transcripts. Intron locations are generally conserved within sequences from different Kingdoms, but not between Kingdoms; many Protista sequences do not have introns in this region. (**C**) Maximum likelihood phylogenetic tree of unicellular eukaryotic GRLs, and selected animal GRLs and plant DUF3537 proteins, with aBayes branch support values. Although the tree is represented as rooted, the rooting is highly uncertain. Protein labels are in black for animals, orange for fungi, blue for protists and green for plants. The scale bar represents one substitution per site.

The online version of this article includes the following figure supplement(s) for figure 2:

**Figure supplement 1.** Alignment of GRL superfamily members.

**Figure supplement 2.** Phylogenetic tree derived from a trimmed multiprotein alignment.

and plant DUF3537 proteins highlights the absence of any universally conserved residues, beyond the hydrophobic regions predicted to be TM domains (*Figure 2—figure supplement 1*). These TM regions are most confidently aligned in the C-terminal halves of the proteins, with substantial variation in loop lengths fragmenting the N-terminal halves (*Figure 2—figure supplement 1*). This pattern of conservation along the protein length is characteristic of insect OR/GR families (*Robertson, 2015*; *Robertson, 2019*; *Saina et al., 2015*), which might reflect the role of the N-terminal half in ligand-recognition (and commensurate higher divergence between proteins) and the C-terminal half in mediating subunit interactions and forming the ion channel pore (*Butterwick et al., 2018*).

To gain an initial idea of the phylogeny of these proteins, we inferred a maximum likelihood tree (*Figure 2C*). As expected, the animal and plant/algal proteins form distinct clades. The sets of GRLs of *V. brassicaformis* and *T. trahens* GRLs segregate into two lineages, one of which (comprising

VbraGRL1/2 and TtraGRL1/2/3) is more closely related to animal GRLs, while the others are more distantly related; this distinction matches observations from the alignments of HMMs derived from these sequences (*Figure 1—figure supplement 1*). Low branch support does not allow for a confident placement of the fungal GRLs; they could group with either of the two protist lineages. All of these observations are consistently held if the tree is inferred from trimmed alignments (*Figure 2—figure supplement 2*).

## Common three-dimensional structural predictions of GRLs and DUF3537 proteins

To obtain further evidence supporting the homology of these proteins, we performed *ab initio* structure predictions of animal GRLs (including AbakORCO as control), plant DUF3537 proteins and unicellular eukaryotic GRLs, using the transform-restrained Rosetta (trRosetta) algorithm (*Yang et al., 2020*). Most query sequences successfully seeded a multisequence alignment to permit extraction of co-evolutionary couplings, generation of inter-residue contact maps and prediction of three-dimensional models (*Figure 3A–B*, *Supplementary file 7*; full outputs of modeling are provided in the Dryad repository doi:10.5061/dryad.s7h44j15f). The contact maps predicted consistent patterns of anti-parallel packing of TM helices (*Figure 3A*). Concordantly, the top-predicted models were qualitatively similar to the AbakORCO cryo-EM structure (*Figure 3B*), particularly in the transmembrane core of these models (*Figure 3B*). Importantly, the consistent helical packing of GRLs and DUF3537 proteins is fundamentally different to an unrelated seven TM protein, the *Homo sapiens* Adiponectin Receptor 1 (HsapAdipoR1), which – despite sharing the same membrane orientation as GRLs – displays an arrangement of helices that is convergent with that of GPCRs (*Hopf et al., 2015*; *Vasiliauskaité-Brooks et al., 2017*; *Figure 3B*). We confirmed these observations first by heuristic searches of the Protein Data Bank (PDB) with Dali (*Holm and Rosenström, 2010*): strikingly, essentially all models identified the AbakORCO cryo-EM structure as the top hit (*Supplementary file 7*; outputs of Dali searches are provided in the Dryad repository doi:10.5061/dryad.s7h44j15f). Second, quantitative pairwise comparisons of selected structures using both Dali and TM-align (*Zhang and Skolnick, 2005*) indicated that all family members are likely to adopt the same global protein fold, while displaying only random structural similarity to our negative control HsapAdipoR1 (*Figure 3C*).

We extended this analysis by generating protein models with an independent algorithm, RaptorX (*Källberg et al., 2012*), and assessing these using Dali PDB searches. Where reliable models were generated (excluding those resulting from multisequence alignments consisting of only a few proteins), AbakORCO was again repeatedly identified as the top hit for animal and unicellular eukaryotic GRLs and plant DUF3537 proteins (*Supplementary file 7*; full outputs of modeling are provided in the Dryad repository doi:10.5061/dryad.s7h44j15f).

It is important to recognize that the trRosetta and RaptorX models of unicellular eukaryotic GRLs derive from multisequence alignments containing large numbers of animal proteins. As such, construction of these models necessarily depends upon amino acid covariation within the animal GRL family. At the level of the global fold, this is only problematic if the query sequences are not homologous to the other sequences in the alignment, a possibility that is inconsistent with our primary and secondary sequence similarities, HMM alignments and phylogenetic analysis.

Importantly, the models of the plant proteins used information extracted from alignments of <u>only</u> other DUF3537 family members, presumably because the primary sequence similarity of members of the more divergent plant and animal proteins is below the threshold of both trRosetta and RaptorX algorithms. The observation that the independently-generated plant protein models are also similar to AbakORCO argues that, despite their high sequence divergence, animal and protist GRLs and plant DUF3537 proteins all adopt a common three-dimensional architecture.

## Discussion

Claims of evolutionary relationships between proteins whose sequence identity resides in the 'twilight zone' must be made with caution (*Rost, 1999*). Nevertheless, our primary, secondary and tertiary structural analyses together support the hypothesis that animal ORs/GRs/GRLs, newly-recognized unicellular eukaryotic GRLs, and plant DUF3537 proteins are homologous. The extremely sparse phylogenetic distribution of these genes in unicellular eukaryotes, despite our efforts to perform exhaustive searches, has multiple potential explanations. These genes might have been

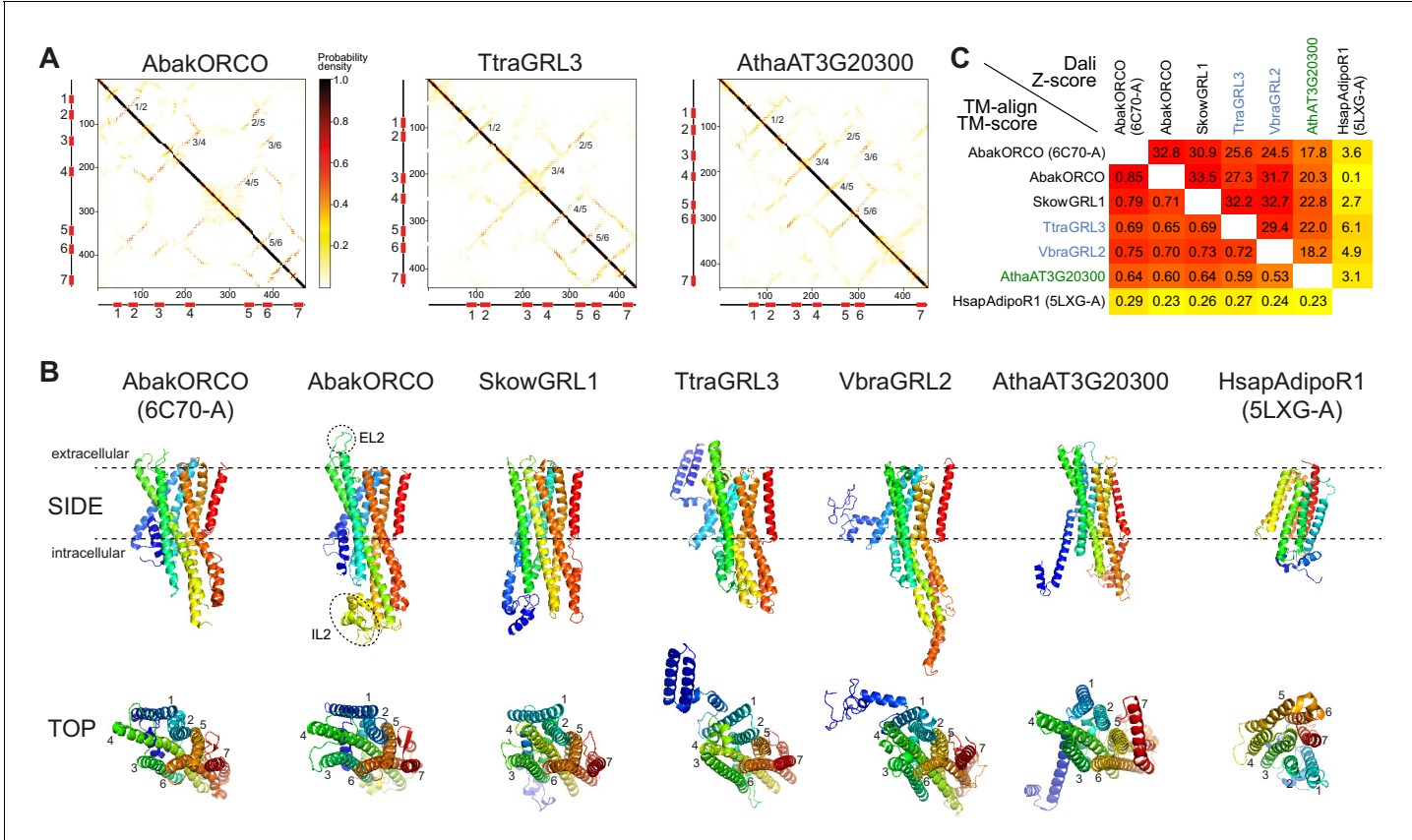

**Figure 3.** *Ab initio* structural predictions of GRLs and DUF3537 proteins. (**A**) Inter-residue contact maps from trRosetta analysis of the indicated proteins. The axes represent the indices along the primary sequence; the positions of the predicted TM domains are shown in the schematics. The representation is mirror-symmetric along the diagonal; in one half 'lines' of contacts perpendicular to the diagonal of the map support the existence of anti-parallel alpha-helical transmembrane packing arrangements. Most pairs of predicted anti-parallel TMs are conserved across the proteins, despite variation in the length of loops between TM domains, supporting a globally similar packing of TM helices. The output of trRosetta analyses for these and other proteins is summarized in *Supplementary file 7* and complete datasets are provided in the Dryad repository (doi:10.5061/dryad.s7h44j15f). (**B**) Side and top views of experimentally-determined (AbakORCO (PDB 6C70 chain A)) and *Homo sapiens* Adiponectin Receptor 1 (HsapAdipoR1; PDB 5LXG chain A [*Vasiliauskaité-Brooks et al., 2017*]) or the top trRosetta protein model of GRL and DUF3537 proteins. All GRL/DUF3537 proteins have a similar predicted global packing of TM domains (which is particularly evident in the top view in which the seven TM domains are labelled), despite variation in lengths of the loops and N-terminal regions (colored in dark blue). By contrast, HsapAdipoR1 has a fundamentally different arrangement of TM domains. The dashed ovals on the AbakORCO model highlight the extracellular loop 2 (EL2) and intracellular loop 2 (IL2) regions that were not visualized in the ORCO cryo-EM structure (*Butterwick et al., 2018*). (**C**) Quantitative pairwise comparisons of the structures shown in (**B**) using TM-align (*Zhang and Skolnick, 2005*) and Dali (*Holm and Rosenström, 2010*). TM-scores of 0.0–0.30 indicate random structural similarity; TM-scores of 0.5–1.00 indicate that the two proteins adopt generally the same fold (1.00 represents a perfect match). Dali Z-scores of <2 indicate spurious similarity. In both cases, these quantitative cut-offs are not stringent, and must be used as a guide in combination with other criteria (e.g., evidence for homology based upon primary sequence comparisons). The two half-matrices are colored using different scales.

independently lost in many lineages and/or have diverged beyond sequence-based homology detection levels. It is also possible that some homologs were acquired by lateral gene transfer, as has been proposed to explain the patchy phylogenetic distribution of microbial rhodopsins (*Gavelis et al., 2017*).

The global conservation of the structural features of unicellular eukaryotic GRLs and DUF3537 proteins with insect chemosensory receptors suggests that they are also ligand-gated ion channels. There is currently insufficient knowledge of the biology of the unicellular eukaryotes in which GRLs have been found – let alone what might be common to these species – to predict ligands or physiological functions. It is possible, if not likely, that the proteins fulfill distinct roles in different phyla, as has been suggested in non-Bilateria, where some GRLs appear to act during development (*Saina et al., 2015*). In *A. thaliana*, transcriptomic analysis indicates that DUF3537 genes have

various tissue-specific expression patterns, including in leaf guard cells, roots and pollen grains (*Schmid et al., 2005*). For one broadly expressed plant protein (AT4G22270), GFP-tagging revealed localization in the plasma membrane (*Guan et al., 2009*), and transgenic overexpression appeared to promote organ growth (*Guan et al., 2009*). However, the significance of this phenotype will require validation by loss-of-function genetic analysis. Together, the available evidence indicates that even if members of this superfamily have a common function as ligand-gated ion channels, they are likely to recognize very diverse chemicals that are potentially of environmental and/or internal origin.

While functional studies remain a future challenge, the recognition of their existence across Animalia, Plantae, Fungi and Protista provides evidence that this protein superfamily originated in the last common eukaryotic ancestor, 1.5–2 billion years ago (*Hedges et al., 2006*). No sequences bearing resemblance to GRLs were found, so far, in Bacteria or Archaea. Future analysis of additional genomic data will help to update the present survey and refine (or refute) the current evolutionary model.

## Materials and methods

### Identification and assessment of candidate GRL homologs

Candidate GRL sequences from unicellular eukaryotes were initially identified by searches of the GenBank RefSeq non-redundant protein sequence database (which includes sequences from 573 protozoan and 13,970 fungal species). The iterative search algorithms PSI-BLAST (*Altschul et al., 1997*) and DELTA-BLAST (*Boratyn et al., 2012*) were used, with a range of divergent animal GRLs (Pfam 7tm_6 (PF02949) and 7tm_7 domain (PF08395)) and plant DUF3537 proteins (PF12056) as queries. To avoid convergence onto hits from Animalia or Plantae, sequences from Metazoa (taxid:33208) or Viridiplantae (taxid:33090) were generally excluded from the search set. We retrieved sequences that had a query coverage of >50% and an E-value of <0.05. Sequences were subject to initial assessment based on their fulfillment of most or all of the following properties: (i) reciprocal BLASTP (or PSI-BLAST) with the candidate as query of Metazoan and Plantae datasets identified a known GRL or DUF3537 sequence as a top hit and/or no significant similarity to other protein families (e.g., distinct types of ion channels or transporters); (ii) the candidate sequence is ~350–500 amino acids long, similar to the vast majority of GRLs and DUF3537 proteins (the TtraGRL3 accession (XP_013759733.1) is 1014 amino acids but this was reannotated to 440 amino acids by trimming a large C-terminal region, which is encoded by the exon of a 3' gene); (iii) the candidate sequence is predicted to contain seven TM domains, an intracellular N-terminus, and generally longer intracellular loops than extracellular loops (as described for insect ORs [*Otaki and Yamamoto, 2003*]; note this analysis was published before recognition of the inverted topology of the insect proteins). Membrane topology predictions were made with both the TMHMM Server v2.0 (*Krogh et al., 2001*; *Figure 1B*) and TOPCONS (*Bernsel et al., 2009*; *Supplementary file 2*). As described previously for Arthropoda ORs/GRs and animal GRLs (*Saina et al., 2015*), TM domains are not always reliably predicted, so visual inspection of the output plots was essential to recognize hydrophobic regions that fell below the threshold for TM domain assignment. Conversely, an N-terminal helical region that forms a re-entrant loop in ORCO (*Butterwick et al., 2018*) was often mispredicted to be a TM domain (see *Figure 1B* legend).

Retained hits from unicellular eukaryotes were used as queries in further PSI-BLAST/DELTA-BLAST searches, as well as in TBLASTN searches (*Gertz et al., 2006*) of the corresponding species' genomes to identify unannotated protein coding sequences (the latter approach ultimately found none in the present analysis). New candidate sequences were subject to the same assessment as described above. Although our searches were very broad phylogenetically, the extreme divergence in the primary sequence of these proteins and the relatively stringent criteria for retaining hits – to avoid excessive numbers of spurious matches with other polytopic membrane proteins – make it highly likely that additional members of the family exist.

To obtain further evidence for homology between the identified sequences, HMMs were constructed with HHblits with default parameters (*Remmert et al., 2012*), using three iterations over the Uniclust30 database (*Mirdita et al., 2017*; *Steinegger and Söding, 2017*). The results of these iterative searches were examined to verify that no additional candidate GRL sequences had been

identified (*Supplementary file 3*). The HMMs resulting from the iterative searches were aligned pairwise using HHblits to obtain a matrix of homology probabilities (code provided in *Supplementary file 4*).

Intron positions were identified by analysis of the predicted gene structure of the coding sequence of each GRL, which was obtained from the corresponding GeneID page for each GenBank Accession.

### Phylogenetic analysis

The multiprotein alignment was built with MAFFT v7.310 (option 'linsi') (*Katoh and Standley, 2013*), and a maximum likelihood tree was inferred with IQTree v.2.0.6 (*Minh et al., 2020*) with aBayes branch support values (*Anisimova et al., 2011*). Alignment trimming for *Figure 2—figure supplement 2* was performed with trimAl (option 'gappyout') (*Capella-Gutierrez et al., 2009*). Raw untrimmed and trimmed sequence alignments are provided in *Supplementary files 5–6*. Alignments were visualized in Jalview 2.9.0b2 (*Waterhouse et al., 2009*) and trees in phylo.io (*Robinson et al., 2016*).

### Protein structure prediction

*Ab initio* protein structure prediction was performed using trRosetta (https://yanglab.nankai.edu.cn/trRosetta/) (*Yang et al., 2020*) and RaptorX (http://raptorx.uchicago.edu/ContactMap/) (*Källberg et al., 2012*). For both algorithms, individual sequences of unicellular eukaryotic GRLs, plant DUF3537 proteins and selected animal GRLs were provided as queries. The complete outputs of the analyses are provided in the Dryad repository (doi:10.5061/dryad.s7h44j15f), and summarized in *Supplementary file 7*. Models predicted for sequences that only seeded multiprotein sequence alignments containing very few (<15) sequences or, for trRosetta models, those with an 'estimated TM-score' of <0.17 (*Yang et al., 2020*) (indicating spurious structural models [*Yang et al., 2020*; *Zhang and Skolnick, 2004*]), were not analyzed further. The top predicted models from trRosetta (model 1) and RaptorX (the model with the lowest estimated RMSD) were used in heuristic Protein Data Bank (PDB) searches with Dali (*Holm and Rosenström, 2010*). The top hits of these searches are shown in *Supplementary file 7* and full search results are provided in the Dryad repository (doi:10.5061/dryad.s7h44j15f).

Pairwise structural similarities of selected trRosetta-predicted models were assessed with TM-align (*Zhang and Skolnick, 2005*) and Dali (*Holm and Rosenström, 2010*), including the experimentally-determined ORCO structure (PDB 6C70) (*Butterwick et al., 2018*) and, as 'negative' control, the *Homo sapiens* Adiponectin Receptor 1 structure (HsapAdipoR1; PDB 5LXG chain A [*Vasiliauskaité-Brooks et al., 2017*]). Models were visualized in PyMol v2.4.0.

## Acknowledgements

We thank members of the Benton lab for discussions. Research in RB's laboratory is supported by the University of Lausanne, European Research Council Consolidator and Advanced Grants (615094 and 833548), the Swiss National Science Foundation (31003A_166646) and the Novartis Foundation for medical-biological Research. DM and CD were supported by Swiss National Science Foundation Grant 183723.

## Additional information

### Funding

| Funder | Grant reference number | Author |
| --- | --- | --- |
| H2020 European Research Council | 833548 | Richard Benton |
| FP7 Ideas: European Research Council | 615094 | Richard Benton |
| Novartis StiftungfürMedizinisch-Biologische Forschung | | Richard Benton |

| Schweizerischer Nationalfonds zur Förderung der Wissenschaftlichen Forschung | 31003A_166646 | Richard Benton |
| Schweizerischer Nationalfonds zur Förderung der Wissenschaftlichen Forschung | 183723 | Christophe Dessimoz David Moi |

The funders had no role in study design, data collection and interpretation, or the decision to submit the work for publication.

## Author contributions

Richard Benton, Conceptualization, Data curation, Supervision, Funding acquisition, Validation, Investigation, Visualization, Writing - original draft, Project administration, Writing - review and editing; Christophe Dessimoz, Data curation, Supervision, Funding acquisition, Validation, Investigation, Visualization, Writing - review and editing; David Moi, Data curation, Software, Validation, Investigation, Visualization, Writing - review and editing

## Author ORCIDs

Richard Benton (iD) https://orcid.org/0000-0003-4305-8301
Christophe Dessimoz (iD) http://orcid.org/0000-0002-2170-853X
David Moi (iD) https://orcid.org/0000-0002-2664-7385

## Decision letter and Author response

Decision letter https://doi.org/10.7554/eLife.62507.sa1
Author response https://doi.org/10.7554/eLife.62507.sa2

## Additional files

### Supplementary files

• Supplementary file 1. Protein sequences of candidate unicellular eukaryotic GRLs. Provisional protein nomenclature (as used in the figures) is indicated in the header of each sequence. Note that these names do not imply orthology between species. Manual corrections to sequences are also noted in the header (e.g., in TtraGRL3 a large C-terminal region was removed as this is likely due to an incorrect merging of exons of adjacent genes that are separated by a gap in the genomic sequence assembly).

• Supplementary file 2. TOPCONs analysis output of candidate unicellular eukaryotic GRLs.

• Supplementary file 3. Sequences retrieved through the HMM searches. Each row contains a separate hit, with identifier, probability, length, query, and score.

• Supplementary file 4. Code to generate *Figure 1—figure supplement 1* (matrix of pairwise HMM alignment probabilities). The Python code is provided as a Jupyter notebook in HTML format, and includes the specific arguments used to run the HHsuite tools.

• Supplementary file 5. Multiprotein alignment of GRLs and DUF3537 proteins.

• Supplementary file 6. Trimmed multiprotein alignment of GRLs and DUF3537 proteins.

• Supplementary file 7. *Ab initio* protein modeling of GRLs and DUF3537 proteins. Results from trRosetta and RaptorX analyses with the indicated protein queries. Multiple sequence alignments (MSAs) were built automatically with the indicated numbers of sequences. In several cases, insufficient sequences were aligned, leading to inadequate data for prediction of inter-residue contacts. For these proteins, the 'estimated TM-scores' of the trRosetta models are commensurately low (scores < 0.17 are likely to reflect spurious protein structural models [*Yang et al., 2020*; *Zhang and Skolnick, 2004*]) and further analysis was not pursued (as indicated by the grey cells). For the other proteins, the top hit and corresponding Z-score from Dali searches of the full Protein Data Bank (PDB) with the top-predicted model are shown. The top model for RaptorX output was defined as that with the lowest 'estimated RMSD' (root-mean-squared deviation, i.e., the estimated average distance deviation (in Å) of the model from the real structure). For almost all models, individual chains

of the AbakORCO homotetramer (A-D) were retrieved as top hits, with a much higher Z-score than the next, non-ORCO hit. TtraGRL4 and TtraGRL5 models were built using MSAs containing only a subset of plant DUF3537 proteins. Although the trRosetta estimated TM-scores were above the threshold (i.e.,>0.17), the retrieved Dali top hits did not have stand-out Z-scores and are likely to be spurious (DIABLO is a HECT-type E3 ligase and PLECTIN is a cytoskeletal protein); here the Z-score of the AbakORCO hit is also given. Dali searches with several of the RaptorX models of the plant proteins identified de novo designed (i.e., artificial) proteins or completely unrelated molecules as the top hit, but AbakORCO was usually also retrieved, with a lower Z-score, as indicated. Full output of trRosetta and RaptorX analyses and Dali searches are provided in the Dryad repository (doi:10.5061/dryad.s7h44j15f).

- Transparent reporting form

### Data availability

All data generated or analyzed during this study are included in the manuscript and supporting files. The output of trRosetta and RaptorX analyses and subsequent Dali searches are available on Dryad (https://doi.org/10.5061/dryad.s7h44j15f).

The following dataset was generated:

| Author(s) | Year | Dataset title | Dataset URL | Database and Identifier |
|---|---|---|---|---|
| Benton R, Dessimoz C, Moi D | 2020 | A putative origin of the insect chemosensory receptor superfamily in the last common eukaryotic ancestor | https://doi.org/10.5061/dryad.s7h44j15f | Dryad Digital Repository, 10.5061/dryad.s7h44j15f |

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
