## [Decision Letter]

**Acceptance summary:**

The paper provides significant insights into the superfamily of olfactory receptors and how it evolved: Your discovery of GRLs in multiple unicellular organisms supports the claim that this is a very old family, even if the sequence conservation is pretty low. However, a major advance results from your analysis of the tertiary structure of these proteins that takes advantage of the power of Rosetta to provide evidence that the GRL proteins are distant members of the same superfamily. This represents a significant advance in our understanding of the origins of this superfamily of proteins.

**Decision letter after peer review:**

Thank you for submitting your article "A putative origin of insect chemosensory receptors in the last common eukaryotic ancestor" for consideration by *eLife*. Your article has been reviewed by four peer reviewers, and the evaluation has been overseen by a Reviewing Editor and Piali Sengupta as the Senior Editor. The reviewers have opted to remain anonymous.

The reviewers have discussed the reviews with one another and the Reviewing Editor has drafted this decision to help you prepare a revised submission.

Summary:

The reviewers found that the paper provides significant insights into this family of receptors: First, your discovery of GRLs in multiple unicellular organisms supports the claim that you are dealing with a large family with plant homologs, although the analyses of sequence conservation remains speculative. However, the major advance results from the tertiary structures of these proteins that take advantage of the power of trRosetta to provide evidence that the GRL proteins are distant members of the same superfamily. This represents a significant advance in our understanding of the origins of this superfamily of proteins.

However, the reviewers had also two major concerns: One is the serious lack of technical details and you must provide more information about how many genomes were used in your initial search and discuss whether it was exhaustive or so stringent that more members of the family likely exist: Providing more technical details will help make the work more accessible. The second point is that functional data would be very useful, e.g. showing biochemically that distant members behave similarly to the fly proteins, or that they serve (or not!) as ligand-gated channels. If you have already acquired this type of data, they would strengthen your paper. However, a discussion of possible molecular functions would be sufficient in the absence of such data.

Reviewer #1:

Vertebrate and nematode odorant receptors (ORs) function as GPCRs, while insect ORs were derived from gustatory receptors (GRs) and function as ligand gated ion channels. However, the evolutionary origin of insect GRs is not clear. The manuscript of Benton, Dessimoz and Moi titled "A putative origin of insect chemosensory receptors in the last common eukaryotic ancestor" answered this key question. Following the previous studies that identified GR-like proteins (GRLs) in animals, and GR homologs, known as the DUF3537 domain-containing proteins in plants, they further identified and performed phylogenetic analysis on GRL proteins in unicellular eukaryotic organisms, including fungi, protists, and algae, the common ancestor of plants and animals.

Overall, the topic of this manuscript is very interesting and well written. The data are solid. Several key points have been addressed, including role of TM7, consistent predicted orientation of TM domains, presence of intracellular loops (like ORCO), conserved vs diverse regions on GRL proteins, and same origin for plant and animal GRLs. Therefore, I strongly recommend for publication, after the authors properly address the following concerns:

1) The major weakness is that there is no functional analysis. If any of GRL proteins is predicted to be a canonical chemical sensor, would it be possible to utilize *Xenopus* or another system to test the hypothesis?

2) If functional study is currently a big challenge, could the authors perhaps add some validation on GRL protein localization in a unicellular eukaryote? I wonder if antibody could be made and used to test membrane localization of GRL, or a tagged protein could be ectopically expressed in a cell line (or yeast).

3) "heteromeric (probably tetrameric) complexes composed of a tuning OR, which recognises odour ligands, and a universal co-receptor, ORCO" This describes a dimeric complex with one OR and one ORCO. It seems not consistent with "probably tetrameric"

4) Introduction paragraph three provides examples of non-chemosensation functions of GRL proteins. I suggest to expand and add a table or a supplemental table, which should include currently known expression patterns and functions of GR and GRL proteins in animals and plants.

Reviewer #2:

In this work, Benton and colleagues consider the evolutionary origin of the immense insect chemoreceptor family, which includes odorant receptors (ORs) and gustatory receptors (GRs). Past sequence mining from the Benton lab and others has suggested that distant members of the GRL family were found in diverse Protostomia and also homologous to a family of uncharacterized plant proteins containing the Domain of Unknown Function 3537. However, despite multiple GRL lineages being present in early branching deuterostomes, GRLs have been completely lost from the chordate lineage suggesting recurrent independent losses, obscuring their exact evolutionary trajectory. Here Benton and colleagues extend their genome mining analyses to identify 17 sequences from fungi, protista and unicellular plants that share the same overall topology and some of the poorly conserved sequence features of this family. Finally, they use the extraordinary power of trRosetta to predict candidate GRL structures from the diverse lineages de novo and demonstrate that they share the same distinct architecture as an experimental structure of an OR. By far the most impressive part of the manuscript is the structure prediction since it would argue that these distantly related members, even bearing little sequence conservation, fold into the same distinct helical arrangement. If correct, this would argue that the GRL family is incredibly ancient, originating in the last eukaryotic ancestor, 1.5-2 Billion years ago, which has important implications for thinking about how this immense family arose.

Overall, I have a few concerns that should be addressed:

1) The Materials and methods are quite sparse and require a lot of effort by the reader to appreciate how well controlled and vetted their results are. Only 17 members of the family were found across the genomes of fungi, protista and unicellular plants, derived from an even smaller subset of species, which the authors acknowledge is extremely sparse and implies either that they propagated by lateral gene transfer or were independently lost many times, making their evolutionary origin still a bit uncertain. The authors should provide more information about how many genomes were used in their initial search and discuss whether it was exhaustive or so stringent that more members of the family likely exist.

2) One complication of the limited number of sequences from unicellular eukaryotes is that the structure prediction relies on multiple sequence alignments largely built from GRs. This was not obvious from the Materials and methods. I only know this because I took one of their putative GRL sequences and submitted it to the trRosetta website and three hours later got the same structure prediction as in Figure 3 and the MSA the trRosetta algorithm used for prediction. While the algorithm for trRosetta has been previously published, for a general audience the paper would benefit from more detail about how it was used-both what was required as input (apparently just a single sequence plugged into the trRosetta website) and how to evaluate the output, beyond physical inspection. For example, in Figure 3C the assignment of proteins to their groups seems like an arbitrary delimitation without further explanation, since the score/distances between proteins are marginally different. Only in the figure legend it states: TM-scores of 0.0-0.30 indicate random structural similarity; TM-scores of 0.5-1.00 indicate that the two proteins adopt generally the same fold. The authors thus suggest a TM score of 0.27 as meaning Orco and HsapAdipoR1 are unrelated but a score of 0.53 as being indicative that VbraGRL2 and AthaAT3G20300 are part of the same structural family, but provide insufficient information to the reader to understand whether this is a stringent cutoff or not.

3) One important caveat that the authors should discuss and address is that given that the de novo structure prediction relies heavily on GR sequence covariation, is there any possibility that tertiary structural similarity is imposed onto these more distant members of the GRL family? Ideally the de novo structure prediction would be truly independent and based on similar numbers of GRL sequences from single-celled eukaryotes but this does not seem possible.

4) The central advance of this study over past work from the Benton lab (Benton, 2015; Hopf et al., 2015) is the dramatic improvement in structure prediction algorithms, which provide tantalizing information about structural similarity (barring the caveat in the point directly above.) I appreciate that the authors don't overstate their claims, suggesting that these GRL proteins may not serve the same function in different organisms but likely form ligand gated channels. To really move into novel territory, I wish the authors could probe the functional or biochemical properties of these ancient GRLs a bit further. For example, for these proteins to serve as ion channels likely requires a multimeric organization. Native gels could biochemically demonstrate this, providing powerful additional evidence that these are part of the same family. Alternatively, could sequence covariation provide evidence for this (e.g. Hopf, 2014). Either way, it would be valuable to discuss this additional feature that does not immediately fall out of the trRosetta predictions.

Reviewer #4:

Benton et al. is a well written study on the evolution of insect chemosensory receptors that uses bioinformatics-based approaches to identify putative GRL homologs in several species of unicellular eukaryotes. Both sequence and structure-based approaches are utilized to buttress the authors arguments that fungal and protista GRL homologs are an evolutionary link to DUF3537 proteins they have previously identified in plants and algae thereby extending this evolutionary relationship to "the last common eukaryotic ancestor"

While I am generally supportive of the authors rationale and recognize they have been careful to appropriately qualify their hypothesis throughout this work, I am somewhat disinclined to place a high degree of definitive value on the ab initio structural predictions which underscores much of this analysis. Even so, and despite the fact these evolutionary relationships between animal and plant GRLs are unlikely to ever be definitively tested, this hypothesis seems to me to be reasonable. That said, I remain underwhelmed by their significance.

Reviewer #5:

The insect chemoreceptor superfamily of ligand-gated ion channels is one of the largest and most diverse protein families known. Partly as a result of their extreme divergence, the evolutionary origins of the superfamily have been obscure. Following up on a previous proposal of relationship to a protein family that is widespread in plants, the authors discovered several convincingly related proteins encoded by fungal, protist, and algal genomes. While the relationship with the plant protein family remains remarkably distant, their three-dimensional modeling of these diverse proteins reveals convincing similarity and hence suggests the superfamily originated at or before the eukaryotic origin.

I have no substantive concerns. Previous objections to the distant relationship to the plant protein family on the basis of lack of three shared introns are not decisive given the rampant loss of introns in the unicellular genomes examined here. The details of the three-dimensional modeling are not my expertise, however these authors previously employed a related technique to generate a remarkably good model for the insect odorant receptors that was mostly confirmed by subsequently generated experimental structure.

---

## [Author Response]

Summary:The reviewers found that the paper provides significant insights into this family of receptors: First, your discovery of GRLs in multiple unicellular organisms supports the claim that you are dealing with a large family with plant homologs, although the analyses of sequence conservation remains speculative. However, the major advance results from the tertiary structures of these proteins that take advantage of the power of trRosetta to provide evidence that the GRL proteins are distant members of the same superfamily. This represents a significant advance in our understanding of the origins of this superfamily of proteins.However, the reviewers had also two major concerns: One is the serious lack of technical details and you must provide more information about how many genomes were used in your initial search and discuss whether it was exhaustive or so stringent that more members of the family likely exist: Providing more technical details will help make the work more accessible.

We acknowledge this concern and have now provided additional technical details on the initial searches and other analyses in the Materials and methods. We further note that all code and sequence files are provided as Supplementary files, and outputs of the *ab initio* protein modelling are available on the Dryad repository (doi:10.5061/dryad.s7h44j15f).

We hope these efforts will clarify the search strategies taken and aid in the reproduction and extension of this work by others. Although our searches have been very broad phylogenetically, the extreme divergence in the primary sequences of these proteins and the relatively stringent criteria for retaining hits – to avoid excessive numbers of spurious matches with other polytopic membrane proteins – make it highly likely that additional members of the family exist (as we now stress in the Discussion and Materials and methods sections). In this work, we have preferred to be relatively conservative by including proteins for which several lines of evidence support their homology to insect chemosensory receptors (i.e., from amino acid sequence similarity and predicted secondary and tertiary structural analyses). Although finer scale details of the evolution of this superfamily will likely emerge in the future, we believe the current data support the central conclusion of our work (i.e., the origin of the insect chemosensory receptor superfamily in the last common eukaryotic ancestor).

The second point is that functional data would be very useful, e.g. showing biochemically that distant members behave similarly to the fly proteins, or that they serve (or not!) as ligand-gated channels. If you have already acquired this type of data, they would strengthen your paper. However, a discussion of possible molecular functions would be sufficient in the absence of such data.

We also would very much like to have functional data on these phylogenetically distant homologs, but do not have anything to add to the current manuscript. Functional characterization is far from trivial: if they are ion channels, it is unknown what ligands might gate them; if they are not channels, it is not obvious how to determine what biochemical function(s) they do possess. Our planned initial approach would be reverse genetic; while this is certainly conceivable for the plant proteins (using *Arabidopsis thaliana* as a model), for the fungal and protist species possessing GRL homologs, none are yet genetically accessible. Transgenesis was very recently reported in *Spizellomyces punctatus* (Medina et al., *eLife* 2020), raising hope that genome-editing approaches will soon be available in this species.

We have expanded the Discussion to discuss possible molecular functions of family members. While we feel that consideration of roles of unicellular eukaryotic GRLs would be pure speculation at this stage (little is known about the biology of these species), we do incorporate some further information on the plant homologs.

Reviewer #1:Vertebrate and nematode odorant receptors (ORs) function as GPCRs, while insect ORs were derived from gustatory receptors (GRs) and function as ligand gated ion channels. However, the evolutionary origin of insect GRs is not clear. The manuscript of Benton, Dessimoz and Moi titled "A putative origin of insect chemosensory receptors in the last common eukaryotic ancestor" answered this key question. Following the previous studies that identified GR-like proteins (GRLs) in animals, and GR homologs, known as the DUF3537 domain-containing proteins in plants, they further identified and performed phylogenetic analysis on GRL proteins in unicellular eukaryotic organisms, including fungi, protists, and algae, the common ancestor of plants and animals.Overall, the topic of this manuscript is very interesting and well written. The data are solid. Several key points have been addressed, including role of TM7, consistent predicted orientation of TM domains, presence of intracellular loops (like ORCO), conserved vs diverse regions on GRL proteins, and same origin for plant and animal GRLs. Therefore, I strongly recommend for publication, after the authors properly address the following concerns:1) The major weakness is that there is no functional analysis. If any of GRL proteins is predicted to be a canonical chemical sensor, would it be possible to utilize *Xenopus* or another system to test the hypothesis?

As described above in response to the general comments, we also would very much like to have functional data on these phylogenetically distant homologs, but do not have anything to add to the current manuscript. Experimental characterization is far from trivial: if they are ion channels, it is unknown what ligands might gate them (necessitating large-scale chemical screening). If they are not channels, it is unclear how best to determine what biochemical function(s) they do possess. Our planned initial approach would be reverse genetic; while this is certainly conceivable for the plant proteins (using *Arabidopsis thaliana* as a model), for the fungal and protist species possessing GRL homologs, none are yet genetically accessible. Transgenesis was very recently reported in *Spizellomyces punctatus* (Medina et al., *eLife* 2020), raising hope that genome-editing approaches will soon be available in this species.

2) If functional study is currently a big challenge, could the authors perhaps add some validation on GRL protein localization in a unicellular eukaryote? I wonder if antibody could be made and used to test membrane localization of GRL, or a tagged protein could be ectopically expressed in a cell line (or yeast).

While it certainly would be possible to tag these proteins with GFP and express them in a heterologous cell type, we do not think such results alone would be particularly informative. It is almost certain – based upon the secondary structure predictions – that these are integral membrane proteins, but they could potentially localize anywhere within the endomembrane system. Without validation in the endogenous cell types, it would be hard to interpret whether localization patterns are real or artefactual (due to, for example, protein over-expression, an impact of the protein tag or an influence of the heterologous cellular environment). Antibodies might be an alternative tool to assess endogenous protein localization, although there has only been very limited success for generation of effective antibodies against insect receptors; moreover, this approach would require development of immunofluorescence protocols for the fungal or protist species of interest and ideally a means of validating antibody specificity (e.g., by parallel staining of genetic knock-outs of the corresponding GRL).

An early study of one of the plant proteins, *A. thaliana* AT4G22270, revealed that an overexpressed GFP-tagged version displayed membrane localization (Guan et al., 2009). Curiously, this study (mis)predicted the family as having four transmembrane domains and did not recognize the similarity with insect chemosensory receptors. This work also found that overexpression of AT4G22270 led to increases in the size of various plant organs, although the relevance of this phenotype (if any) remains to be confirmed by loss-of-function analysis. Nevertheless, the cellular localization may be real and we cite this work in the revised Discussion.

3) "heteromeric (probably tetrameric) complexes composed of a tuning OR, which recognises odour ligands, and a universal co-receptor, ORCO" This describes a dimeric complex with one OR and one ORCO. It seems not consistent with "probably tetrameric"

We have clarified this sentence to indicate that the tetrameric complex probably comprises two tuning OR subunits and two ORCO subunits.

4) Introduction paragraph three provides examples of non-chemosensation functions of GRL proteins. I suggest to expand and add a table or a supplemental table, which should include currently known expression patterns and functions of GR and GRL proteins in animals and plants.

To our knowledge, the work cited in this paragraph, and the revised Discussion (which incorporates further information on the plant proteins – see the comment above) encompasses all known “non-chemosensory” roles of this family. For completeness, we have now added a sentence to this paragraph on the thermosensory and light-sensing functions of *D. melanogaster* GR28b isoforms. At this stage, we feel that information on non-chemosensory function of members of this repertoire is simply too sparse – and the evidence for certain functions too limited – to warrant a table, which would ultimately be redundant with the information in the text.

Reviewer #2:In this work, Benton and colleagues consider the evolutionary origin of the immense insect chemoreceptor family, which includes odorant receptors (ORs) and gustatory receptors (GRs). Past sequence mining from the Benton lab and others has suggested that distant members of the GRL family were found in diverse Protostomia and also homologous to a family of uncharacterized plant proteins containing the Domain of Unknown Function 3537. However, despite multiple GRL lineages being present in early branching deuterostomes, GRLs have been completely lost from the chordate lineage suggesting recurrent independent losses, obscuring their exact evolutionary trajectory. Here Benton and colleagues extend their genome mining analyses to identify 17 sequences from fungi, protista and unicellular plants that share the same overall topology and some of the poorly conserved sequence features of this family. Finally, they use the extraordinary power of trRosetta to predict candidate GRL structures from the diverse lineages de novo and demonstrate that they share the same distinct architecture as an experimental structure of an OR. By far the most impressive part of the manuscript is the structure prediction since it would argue that these distantly related members, even bearing little sequence conservation, fold into the same distinct helical arrangement. If correct, this would argue that the GRL family is incredibly ancient, originating in the last eukaryotic ancestor, 1.5-2 Billion years ago, which has important implications for thinking about how this immense family arose.Overall, I have a few concerns that should be addressed:1) The Materials and methods are quite sparse and require a lot of effort by the reader to appreciate how well controlled and vetted their results are. Only 17 members of the family were found across the genomes of fungi, protista and unicellular plants, derived from an even smaller subset of species, which the authors acknowledge is extremely sparse and implies either that they propagated by lateral gene transfer or were independently lost many times, making their evolutionary origin still a bit uncertain. The authors should provide more information about how many genomes were used in their initial search and discuss whether it was exhaustive or so stringent that more members of the family likely exist.

As described above in response to the general comments, we acknowledge this concern and have now provided additional technical details on the initial searches and other analyses in the Materials and methods. We further note that all code and sequence files are provided as Supplementary files, and outputs of the *ab initio* protein modelling are available on the Dryad repository (doi:10.5061/dryad.s7h44j15f).

We hope these efforts will clarify the search strategies taken and aid in the reproduction and extension of this work by others. Although our searches have been very broad phylogenetically, the extreme divergence in the primary sequence of these proteins and the relatively stringent criteria for retaining hits – to avoid excessive numbers of spurious hits with other polytopic membrane proteins – make it highly likely that additional members of the family exist (as we now stress in the Discussion and Materials and methods sections). In this work, we have preferred to be relatively conservative by including proteins for which several lines of evidence support their homology to insect chemosensory receptors (i.e., from amino acid sequence similarity and predicted secondary and tertiary structural analyses). Although finer scale details of the evolution of this superfamily will likely emerge in the future, we believe the current data support the central conclusion of our work (i.e., the origin of the insect chemosensory receptor superfamily in the last common eukaryotic ancestor).

2) One complication of the limited number of sequences from unicellular eukaryotes is that the structure prediction relies on multiple sequence alignments largely built from GRs. This was not obvious from the Materials and methods. I only know this because I took one of their putative GRL sequences and submitted it to the trRosetta website and three hours later got the same structure prediction as in Figure 3 and the MSA the trRosetta algorithm used for prediction. While the algorithm for trRosetta has been previously published, for a general audience the paper would benefit from more detail about how it was used-both what was required as input (apparently just a single sequence plugged into the trRosetta website) and how to evaluate the output, beyond physical inspection. For example, in Figure 3C the assignment of proteins to their groups seems like an arbitrary delimitation without further explanation, since the score/distances between proteins are marginally different. Only in the figure legend it states: TM-scores of 0.0-0.30 indicate random structural similarity; TM-scores of 0.5-1.00 indicate that the two proteins adopt generally the same fold. The authors thus suggest a TM score of 0.27 as meaning Orco and HsapAdipoR1 are unrelated but a score of 0.53 as being indicative that VbraGRL2 and AthaAT3G20300 are part of the same structural family, but provide insufficient information to the reader to understand whether this is a stringent cutoff or not.

The reviewer raises a number of important points, which we address individually below:

- *structure predictions from multiple sequence alignments (MSAs) largely built from GRs*: this reviewer reiterates this issue in the comment below, where we provide a full response.

- *use of trRosetta algorithm*: we provide additional use and evaluation of this server in the Materials and methods. In brief, the user interface is indeed extremely simple, requiring just entry of an individual sequence, as MSAs are built automatically.

- *evaluation of trRosetta output*: we describe the pertinent information in Supplementary file 7 and the associated legend. The key parameter to judge the quality of the top model from trRosetta is the “estimated TM-score”. As described in the cited trRosetta paper (Yang et al., 2020), this is calculated based upon a combination of the probability of the predicted top distances and the average pairwise TM-score between the top ten models under no restraints. In test proteins of known structure, the estimated TM-score had a high correlation with the true TM-score (which is calculated based upon comparison of the model and the experimentally-determined protein structure). For proteins for which no experimental information is available (such as GRLs or DUF3537 proteins), the estimated TM-score provides a measure of predicted resemblance of the model to the real structure. While there is no firm cut-off, scores <0.17 are likely to reflect spurious protein structural models (Yang et al., 2020). In our work, as shown in Supplementary file 7, sequences that yielded MSAs with very few proteins gave commensurately extremely low estimated TM-scores (typically around 0.1); these models were not examined further. All trRosetta output files are provided in the Dryad repository (doi:10.5061/dryad.s7h44j15f).

*- evaluation of trRosetta models by structure comparisons with Dali and TM-align*: for all trRosetta models that had an estimated TM-score >0.17, we assessed whether these had similarity to proteins of known structure in the Protein Data Bank using the Dali server. In all but two cases (TtraGRL4 and TtraGRL5), the ORCO cryo-EM structure was identified as the top hit, usually with a Dali Z-score (a measure of structural similarity) that is substantially higher that the next most similar protein fold. The results of these Dali searches are provided inside the corresponding subfolder of the trRosetta output in the Dryad repository. The consistent retrieval of ORCO by other models of animal GRs/GRLs, protist GRLs and plant DUF3537 proteins is striking and argues these proteins all adopt a similar fold. Regarding the two exceptions: the best TtraGRL4 and TtraGRL5 models identified Diablo (a HECT-type E3 ligase) and Plectin (a cytoskeletal protein) as top hits, respectively. Although these GRL models have estimated TM scores >0.17 and the Dali Z-scores are indicative of “significant similarity” (>2 (Holm et al., 2010)), these are clearly spurious matches. We note that in both cases the number of sequences used in the MSA is very low (<230) compared to models of TtraGRL1-3 (>1200).

We further assessed structural similarity by pairwise comparisons of selected proteins (with the highest estimated TM-score) together with a negative control (AdipoR1, which has the same membrane topology as the OR/GR/GRL/DUF3537 superfamily). For Dali pairwise comparisons (top-right of Figure 3C), the Z-score is substantially higher for all comparisons within the OR/GR/GRL/DUF3537 set than with AdipoR1. Similarly, for TM-align pairwise comparisons, the OR/GR/GRL/DUF3537 comparisons all fall within the range of 0.5-1, which indicates – as described in Zhang and Skolnick NAR 2005 – that the proteins are expected to adopt the same fold (1 would be a perfect match). By contrast, comparisons with AdipoR1 fall within the range (0-0.3) indicative of spurious similarity. We tried to add these numerical ranges on the figure itself but found that it cluttered the panel and would prefer to have the full description of their meaning in the legend.

We emphasize that the cut-offs of trRosetta, Dali and TM-align are defined by the developers of these algorithms based upon analysis of many test cases of proteins of known structure. To our knowledge, these cut-offs are not stringent, and must be viewed in the context of the proteins being analyzed, as many factors could impact these scores (e.g., quality of model, quality of experimentally-determined structure, primary sequence similarity target and query, domain organization of protein (in our experience individual proteins with large inserts in the loops were often problematic)). In our work, the tertiary structural similarity provides additional support for the homology between various proteins that were initially identified based upon primary and secondary structural similarities.

To strengthen our claims, we now provide analyses of the same set of query sequences with an independent *ab initio* protein folding algorithm, RaptorX, which uses distance-based protein folding driven by deep learning (Kallberg et al., 2012). While this algorithm failed to build sufficiently large MSAs for slightly more queries than trRosetta, several sequences from both protists and plants successfully yielded models that, via Dali searches, retrieved the ORCO structure as the top hit. The results of this new analysis are summarized in Supplementary file 7, and the complete output files from RaptorX, together with the results of the subsequent Dali searches, are provided in the Dryad repository (doi:10.5061/dryad.s7h44j15f).

We hope to have explained the logic of software use, our steps for quality control at each stage and the availability of the source data to allow readers to view and reproduce our results. As we are users, not testers, of the software packages, we felt it out-of-place to have a detailed description of these published algorithms in our work, but we have added additional technical details in this revision to enable a reader to appreciate our procedures for assessing the structure prediction results.

3) One important caveat that the authors should discuss and address is that given that the de novo structure prediction relies heavily on GR sequence covariation, is there any possibility that tertiary structural similarity is imposed onto these more distant members of the GRL family? Ideally the de novo structure prediction would be truly independent and based on similar numbers of GRL sequences from single-celled eukaryotes but this does not seem possible.

This is a very good point: at present, there are indeed insufficient numbers of GRL sequences from unicellular eukaryotes alone to be able to analyze amino acid co-evolution and use this information for modelling. The current models therefore necessarily depend in part upon covariation within the larger animal GR/GRL family. At the level of the global fold, this is only problematic if the query sequence is not homologous to the sequences in the alignment. We believe that the primary and secondary sequence analyses and phylogenetic analysis (in Figure 1, Figure 1—figure supplement 1, Figure 2) do support such homology, notably for the protist GRLs for which we have obtained structural models.

Importantly, the models of the plant proteins used information extracted from alignments of only other DUF3537 family members, because these are more divergent from the animal sequences than those of unicellular eukaryotes. It is therefore striking that the plant structural models are also similar to ORCO, and infer that the entire family is likely to share the same global fold. We briefly mentioned these issues in our original manuscript but have now expanded our comments on these points in the text.

4) The central advance of this study over past work from the Benton lab (Benton, 2015; Hopf et al., 2015) is the dramatic improvement in structure prediction algorithms, which provide tantalizing information about structural similarity (barring the caveat in the point directly above.) I appreciate that the authors don't overstate their claims, suggesting that these GRL proteins may not serve the same function in different organisms but likely form ligand gated channels. To really move into novel territory, I wish the authors could probe the functional or biochemical properties of these ancient GRLs a bit further. For example, for these proteins to serve as ion channels likely requires a multimeric organization. Native gels could biochemically demonstrate this, providing powerful additional evidence that these are part of the same family. Alternatively, could sequence covariation provide evidence for this (e.g. Hopf, 2014). Either way, it would be valuable to discuss this additional feature that does not immediately fall out of the trRosetta predictions.

As described above in response to the general comments, we feel it is premature to begin to assess biochemical properties of these proteins without first some hint of their in vivo role, which in turn requires genetic analysis. It is currently hard also to extract further insights from patterns of amino acid covariation for the protist and fungal GRLs alone because there are too few sequences available.

We have made some preliminary analysis of the plant proteins, by overlaying the degree of amino acid conservation on the predicted structure but this was not particular informative: in contrast to the animal proteins, the plant family has quite high amino acid identity throughout its length and this analysis did not highlight particularly conserved regions (in 3D space) that might indicate functional domains. Moreover, in contrast to ORs, for which there is good (albeit mostly indirect) evidence of heteromeric complex assembly between tuning ORs and ORCOs, we currently do not know if and how DUF3537 proteins may form multimeric complexes. As it is not trivial to distinguish contacts that may be involved in monomer folding versus those involved in potential intersubunit contacts (as described in Hopf et al., *eLife* 2014), we feel it is premature to attempt to draw conclusions about complex formation from sequence analysis alone at this stage. If such intersubunit interactions exist, we suspect they are slightly different from those reported in ORCO. The cryo-EM ORCO structure revealed that the major interaction interface was within cytoplasmic domain (the “anchor” domain (Butterwick et al., 2018)) comprising cytosolic regions of TM4, TM5, TM6 and TM7a; notably, all of the plant proteins have a cytoplasmic insertion of ∼50 amino acids in this region in IC3 (between TM6 + TM7a).

Reviewer #4:Benton et al. is a well written study on the evolution of insect chemosensory receptors that uses bioinformatics-based approaches to identify putative GRL homologs in several species of unicellular eukaryotes. Both sequence and structure-based approaches are utilized to buttress the authors arguments that fungal and protista GRL homologs are an evolutionary link to DUF3537 proteins they have previously identified in plants and algae thereby extending this evolutionary relationship to "the last common eukaryotic ancestor"While I am generally supportive of the authors rationale and recognize they have been careful to appropriately qualify their hypothesis throughout this work, I am somewhat disinclined to place a high degree of definitive value on the ab initio structural predictions which underscores much of this analysis. Even so, and despite the fact these evolutionary relationships between animal and plant GRLs are unlikely to ever be definitively tested, this hypothesis seems to me to be reasonable. That said, I remain underwhelmed by their significance.

We fully acknowledge the caveats associated with *ab initio* structural predictions and hope to have been suitably cautious in our claims throughout the manuscript. We do find the very strong similarity between the predicted and experimentally-determined ORCO structures striking, which supports the relevance of the predictions for other members of this repertoire. We refer the reviewer to our detailed comments in response to reviewer 2 concerning the procedure, assessment and caveats of *ab initio* modelling, as well as our additional analyses in this revision using the RaptorX algorithm. We provide all the outputs of these analyses in the Dryad repository (doi:10.5061/dryad.s7h44j15f), to permit independent assessment and reproduction by others.